# Non-Curative Treatment Choices in Colorectal Cancer: Predictors and Between-Hospital Variations in Denmark: A Population-Based Register Study

**DOI:** 10.3390/cancers16020366

**Published:** 2024-01-15

**Authors:** Søren Rattenborg, Torben Frøstrup Hansen, Sören Möller, Erik Frostberg, Hans Bjarke Rahr

**Affiliations:** 1Department of Surgery, Vejle Hospital, University Hospital of Southern Denmark, Beriderbakken 4, 7100 Vejle, Denmark; erik.frostberg@rsyd.dk (E.F.); hans.rahr@rsyd.dk (H.B.R.); 2Institute of Regional Health Research, University of Southern Denmark, Campusvej 55, 5230 Odense M, Denmark; torben.hansen@rsyd.dk; 3Colorectal Cancer Center South, Vejle Hospital, University Hospital of Southern Denmark, Beriderbakken 4, 7100 Vejle, Denmark; 4Department of Oncology, Vejle Hospital, University Hospital of Southern Denmark, Beriderbakken 4, 7100 Vejle, Denmark; 5Open Patient Data Exploratory Network, Odense University Hospital, J. B. Winsløws Vej 9A, 3. Sal, 5000 Odense C, Denmark; soren.moller@rsyd.dk; 6Department of Clinical Research, University of Southern Denmark, Campusvej 55, 5230 Odense M, Denmark

**Keywords:** colorectal cancer, oncology, predictors, surgery

## Abstract

**Simple Summary:**

Despite universal free healthcare and national treatment guidelines, between-hospital variation in treatment choices for colorectal cancer has been reported in several countries. It is unknown whether this variation may be ascribed to simple variations in clinical case mix or to differences in socioeconomic status or even attitudes and traditions among patients and healthcare professionals. This study examined factors associated with non-curative treatment choices and with refraining from recommended chemotherapy in colorectal cancer in Denmark in 2009–2018. We found that non-curative surgical treatment was associated with being old and frail and having widespread cancer and low weight. Not having chemotherapy was also associated with previous treatment complications and living alone. Marked variations in non-curative treatment between hospitals were found, even after taking a wide range of plausible explanations into account. The reason for these variations is unknown and requires further examination.

**Abstract:**

Background: Variations in treatment choices have been reported in colorectal cancer (CRC). In the context of national recommendations, we aimed to elucidate predictors and between-hospital variations in refraining from curatively intended surgery and adjuvant chemotherapy in potentially curable colorectal cancer. Methods: A total of 34,116 patients diagnosed with CRC from 2009 to 2018 were included for analyses on non-curative treatment in this register-based study. Subsequently 8006 patients were included in analyses on adjuvant treatment. Possible predictors included patient-, disease-, socioeconomic- and perioperative-related factors. Logistic regressions were utilized to examine the predictors of a non-curative aim of treatment and no adjuvant chemotherapy. Results: The predictors of non-curative treatment were high age, poor performance, distant metastases and being underweight. Predictors for no adjuvant treatment were high age, poor performance, kidney disease, postoperative complications and living alone. For both outcomes we found between-hospital variations to be present. Conclusions: Non-curative overall treatment and refraining from adjuvant chemotherapy were associated with well-known risk factors, but the former was also associated with being underweight and the latter was also associated with living alone. Marked between-hospital variations were found and should be examined further.

## 1. Introduction

The cornerstone in the curative treatment of colorectal cancer (CRC) is surgery. Of the almost 4000 patients diagnosed with CRC in Denmark in 2022, 68% were treated surgically with curative intent [1]. Treatment choices rest on national and European guidelines which have been in place for at least two decades [2,3,4,5], but the final decisions are made together with the individual patient, taking a multitude of patient- and disease-related factors, as well as patient preferences, into account.

The most common reason for refraining from curatively intended surgery is disseminated disease, although palliative surgery may be needed to manage acute problems such as bowel obstruction. Sometimes patients themselves decline curative treatment, and in other cases the surgeon will recommend against major surgery based on an assessment of performance status, age and comorbidity. In these cases, the choice may be compromised oncologic resection or no resection at all [6].

Although the aforementioned characteristics are the major predictors for the non-curative treatment of CRC, factors such as socioeconomic status and psychiatric disease could be important as well, as they are related to adverse outcomes [7,8,9]. Finally, clinical experience, traditions and attitudes may vary between individual surgeons, hospitals, and even countries. Majano et al. examined differences between four Northern European countries in CRC survival and were able to relate them to marked differences in resection rates [10].

The use of adjuvant therapy may also vary. Commonly accepted guidelines recommend adjuvant chemotherapy after curatively intended surgery for Union for International Cancer Control (UICC) stage III cancer [2,3,4]. The evidence of benefit from adjuvant therapy in UICC stage II CRC is more conflicting, but the European Society for Medical Oncology (ESMO) together with the Danish guidelines suggest adjuvant therapy if specific high risk features (e.g., pT4 tumor, fewer than 12 lymph nodes examined in the specimen) are present together with proficient mismatch repair (pMMR) status and favorable performance status [2,3,4]. Present guidelines suggest adjuvant treatment, after relevant risk assessment [2,3,4], consisting of fluoropyrimidines alone or in combination with oxaliplatin. In stage II colon cancer with high risk features, the Danish guidelines suggest fluoropyrimidine monotherapy, while the addition of oxaliplatin is recommended in stage III in patients below the age of 70 [2,3]. For rectal cancer, adjuvant chemotherapy is only recommended in those without neoadjuvant radiochemotherapy [2,4]. Those with upper rectal cancer have the same indications as colon cancer, while for mid and lower rectal cancer fluoropyrimidines as a monotherapy are recommended [2,4]. In certain cases, local radiotherapy can be applied [4].

However, in spite of national guidelines for both the surgical and oncological treatment of CRC, recent annual reports by the Danish Colorectal Cancer Group have found marked between-hospital variations in, e.g., the proportion of patients having surgery for rectal cancer (45–88%) [1] and adjuvant chemotherapy for stage III colon cancer (67–92% and 55–93%) [1,11]. The reason for these variations is unknown, but it may be speculated to reflect the differences in case mix between hospitals that we have shown previously [9].

The aim of this study was to elucidate predictors and between-hospital variations in refraining from curatively intended surgery and adjuvant chemotherapy in potentially curable colorectal cancer.

## 2. Materials & Methods

### 2.1. Study Design and Population

This study is a register-based Danish national cohort study. The data were extracted from the Danish Colorectal Cancer Group (DCCG) database [12], the National Patient Registry (NPR) under the Danish National Health Authority [13] and Statistics Denmark (SD) [13].

The cohort consists of 44,471 adults with a first-time diagnosis of CRC in the years 2009–2018. The cohort and covariates are thoroughly described in Rattenborg et al. [9].

### 2.2. Specific Primary Exclusion Criteria for This Study

In this study we focused on patients with potentially curable disease. Therefore, we added the following exclusion criteria (Figure 1):Incompatibility between the tumor location and resection procedure registered in the DCCG (e.g., primary tumor in the right colon and rectal resection), or synchronous cancer;Patients in whom curative surgery was not an option (e.g., patients who died before surgery or patients not offered surgery because of disseminated disease);Non-elective surgery;Rare and poorly characterized aims of treatment or an unspecified aim of treatment;The presence of distant metastasis (UICC stage IV) was not an exclusion criterion, provided that the surgeon had not registered disseminated disease as the reason for no surgery.

### 2.3. The Stratification of the Cohort by Overall Treatment Goal

Patients were stratified into four different groups based on their intended treatment registered in the DCCG database: operative treatment with curative intent (OT-CUR); operative treatment with compromised or palliative intent (OT-NCUR); non-operative treatment because the patient declined (NOT-NO); and non-operative treatment due to comorbidity (NOT-CO). ‘Compromised surgery’ was defined as a suboptimal oncological resection chosen to minimize the surgical trauma in a frail patient. The ‘palliative surgery’ category had rather few elective patients registered and therefore it was fused with the ‘compromised surgery’ category for our purpose. Data were collected from the DCCG.

### 2.4. Specific Secondary Exclusion Criteria for Adjuvant Oncological Treatment

Patients in the OT-CUR group and with UICC stage II or III disease and with at least one high-risk feature for benefit (inclusion criteria) of adjuvant therapy, according to national recommendations [2,5], were eligible for analysis, if no exclusion criteria were present. Or, in other words, patients in whom adjuvant chemotherapy was indicated according to national guidelines were eligible for analysis. These changed over the decade studied, and for each patient, we applied the criteria in force at the time of that patient’s operation (Appendix A). Patients > 80 years old were excluded, since it was (by the authors) considered a general rule of thumb for no benefit of adjuvant therapy during the whole period. Patients only having local or unspecified resections were excluded from this analysis, as well as (in some years) patients with deficient mismatch repair (dMMR) status, age > 75 years and World Health Organization (WHO) performance status (PS) > 2 (Figure 1 and Appendix A). Recommended adjuvant chemotherapy regimes were, through the whole period, fluoropyrimidine, and possibly leucovorin, and oxaliplatin, depending on the aforementioned risk assessment [2,5]. We grouped included patients as having any or no adjuvant chemotherapy treatment. Data on adjuvant chemotherapy were collected via the DCCG from the NPR.

### 2.5. Predictors and Covariates

#### 2.5.1. Demographics, Lifestyle and Performance Score

Sex, age groups (<50, 50–64, 65–74, 75–84 or ≥85), body mass index (BMI) according to WHO classification (underweight, normal, overweight, obese) [14], alcohol consumption in units per week (0–14, >14 units) and smoking status (never, ex-smoker, current smoker), American Association of Anesthesiologists (ASA) score (I–V, unknown) and WHO performance status (PS) (0–4, unknown) were included as covariates. Data were collected from the DCCG.

#### 2.5.2. Comorbidity

Comorbidity was reported on an overall level as an aggregated Charlson comorbidity index [15] score (0, 1, 2, 3+) in descriptive tables with updated weights [16]. International Classification of Disease 10th edition (ICD-10) codes for CRC were excluded. Only dichotomous comorbidity variables were included in the regression analyses, based on ICD-10 and Anatomical Therapeutic Chemical Classification System (ATC) codes, as reported recently [9]. In brief, the included somatic domains were cardiovascular disease, chronic pulmonary disease, diabetes, dementia, liver disease, kidney disease, chronic nerve disease, other cancer or tumors and connective tissue disease, and the included psychiatric domains were affective disorders, schizophrenia spectrum disorders, disorder of adult personality and behavior and disorders due to psychoactive substance abuse. ICD-10 codes were collected from the NPR and ATC codes from the prescription database at SD.

#### 2.5.3. Socioeconomic Factors

Educational level (short, medium, long, unknown or unclassified) [17], annual household income in Danish kroner (DKK) (1st–4th quartile or unknown) and cohabitation status (cohabitating, alone, unknown) were included. Data were collected from SD.

#### 2.5.4. Disease-Related Factors

The primary tumor was defined as located in either the colon or the rectum. Clinical presentation with distant metastases at the time of diagnosis was included (yes, no, unknown), as well as information on pretreatment discussion at a multidisciplinary team (MDT) conference (yes, no, unknown). The hospital responsible for the definitive treatment was included in the analyses. These hospitals were identified by letters A to Q, ordered by total patient volume (low to high). Data were collected from the DCCG.

#### 2.5.5. Covariates Included Only in Regression Models for Adjuvant Chemotherapy

In addition to the aforementioned variables, the following variables were of interest in the adjuvant chemotherapy analysis: mismatch repair (MMR) status was categorized as proficient MMR (pMMR), deficient MMR (dMMR) or unknown and collected from the DCCG. In the study period, the most common neoadjuvant radio-chemotherapy regime for rectal cancer was 50.4 Gray in 28 fractions and fluoropyrimidine, applied mainly for T4 and T3 (depending on suspected involvement of margins) [2,5]. Neoadjuvant therapy was categorized as no, yes or unknown for those with rectal cancer only and was collected from the DCCG and via the DCCG from the NPR. Data on postoperative medical and surgical complications within 30 days after surgery were included (no, yes or unknown) and collected from the DCCG.

### 2.6. Statistical Methods

Descriptive statistics were applied to examine between-hospital variations in outcome variables. For group comparisons (e.g., between two different proportions) we estimated 95% confidence intervals (CI).

In order to examine predictors for non-curative treatment aims, a multivariable multinomial logistic regression model was utilized. The outcome variable was the aim of treatment with the OT-CUR group as the comparison level. The relevant covariates were included. We applied the Hosmer-Lemeshow test with twenty groups to investigate the goodness of model fit. Results are presented as relative risk ratios (RR) with corresponding 95% CI.

In order to examine predictors for refraining from adjuvant chemotherapy, a multivariable logistic regression model was built for colon and rectal cancer, respectively. The possible outcomes were any kind of adjuvant chemotherapy, with no adjuvant chemotherapy as the comparison level. As mentioned briefly above, exclusion criteria in force at the time of the operation of each particular patient were used to define the population with the indication for adjuvant chemotherapy. Since some of these exclusion criteria may have predicted non-treatment with adjuvant chemotherapy before they were included in national recommendations, we included these criteria in our analysis. To balance the analysis we recoded age, MMR and neoadjuvant therapy to non-applicable (n-a) in the periods when they were exclusion criteria. The exact distribution (without n-a) is reported. Results from this model are presented as odds ratios (OR) with corresponding 95% CI. Sensitivity analyses with a reduction of hospitals to tertiles (low, medium, high) of the total volume of patients were done in order to examine if hospital volume was a predictor.

Missing data were included for all analyses and treated as an unknown level for each variable. We did not perform imputation. Data were stored and managed on the secure servers of Statistics Denmark, using Stata IC/17 (StataCorp LCC, 4905 Lakeway Drive, College Station, TX, USA) for analysis. Data were only extracted after anonymization.

### 2.7. Ethics and Permisson

This study was approved by the DCCG and Danish Clinical Quality Program (DCCG-2018-03-08a) and the Danish Data Protection Agency (jr. no 18/15252). No other approvals were required under Danish law [18].

## 3. Results

For the overall analyses, 34,116 patients were included in this study (Figure 1).

### 3.1. The Characteristics of Overall Treatment Aims

Table 1 shows an overview of the included cohort, stratified by the aim of treatment. The majority (90%) had OT-CUR, with five percent having OT-NCUR. Three percent had NOT-NO and three percent had NOT-CO. During the study period, the proportion of patients receiving curative treatment was consistent. The proportion of patients who had OT-NCUR decreased from eight to three percent over the study period, while proportions of NOT-NO and NOT-CO were relatively constant during the study period.

### 3.2. The Characteristics of Patients with a Curative Treatment Aim

The patients in the OT-CUR group were generally younger, had a lower ASA score and PS, lower Charlson scores, higher educational level, higher annual household income and were more often cohabiting (Table 1). The majority of OT-CUR patients had a segmental resection, while 6% had a local resection (e.g., endoscopic mucosal resection). Also, a few patients (<1%) in this group did not actually receive any curative operative treatment although the intention of the treatment before surgery was curative.

### 3.3. The Characteristics of Patients with a Non-Curative Treatment Aim

The non-curatively treated patients were characterized by higher age, ASA, PS and Charlson score, short educational level and lower annual household income, and were more frequently living alone (Table 1).

OT-CUR and OT-NCUR were more common among patients with colon cancer rather than rectum cancer, while NOT-NO and NOT-CO were less often found in colon cancer patients (Table 1). The most predominant types of surgery in OT-NCUR were resection (47%), relief of obstruction (42%), local resections (7%) or only exploration (4%) (not shown in tables).

Looking at the data descriptively, non-curative treatment varied from 8 to 13% between hospitals as seen in Figure 2. OT-NCUR treatment varied from 2 to 7%. NOT-NO and NOT-CO varied only a few percentages between hospitals, 2 to 4%, and 1 to 4%, respectively.

#### 3.3.1. The Predictors of Non-Curative Treatment in a Multinomial Model

The highest RR values among the predictors of OT-NCUR were distant metastases and advanced age, ASA or PS (Table 2). To a lesser degree BMI, dementia, other cancer disease at diagnosis and rectal location were predictors of OT-NCUR. Between-hospital variations in RR varied significantly, with RR as low as 0.3.

The highest RR among the predictors of both NOT-NO and NOT-CO were high age, ASA and PS (Table 2). Also, rectal cancer, year of diagnosis, distant metastases, BMI and living alone were predictors of both non-operative aims. A range of comorbidities were, not surprisingly, predictors of NOT-CO, while low income was a predictor of NOT-NO. Between-hospital variations in RR shared patterns for NOT-NO and NOT-CO, with RR up to 2.8 for NOT-NO and between 0.3 and 2.1 for NOT-CO. In a sensitivity analysis, where hospitals were contracted to tertiles (1st, 2nd and 3rd) of the total volume of patients, the only outcome which was markedly different with volume was NOT-CO, with a RR for the low volume tertile of 0.61 (95% CI 0.49–0.77) using the high volume tertile as the comparison level (not shown in table).

#### 3.3.2. Adjuvant Chemotherapy

For analyses on adjuvant chemotherapy, 8006 patients with an indication for adjuvant chemotherapy were included (Figure 1). An overview is seen in Table 3. The majority of patients received adjuvant therapy (72%), with the largest proportion in patients with colon cancers. The majority (81%) of UICC stage III received adjuvant treatment, while only 42% of stage II had treatment. Those who did not receive adjuvant chemotherapy were generally of higher age, had a higher ASA and/or Charlson score, had a shorter education, had a lower annual household income, were living alone and had more often had complications within 30 days after surgery. The unadjusted between-hospital variations are seen from Figure 3.

#### 3.3.3. The Predictors of Adjuvant Chemotherapy in a Logistic Regression Model

The multivariable logistic regression tables for having any adjuvant treatment for colon and rectal cancer respectively are shown in Table 4. The most significant predictors associated with not receiving adjuvant therapy (and thereby having a low OR for receiving adjuvant therapy) for both colon and rectum were as follows: high age, ASA III and PS 1 or 2–4, respectively, together with kidney disease, postoperative complications, living alone and earlier years of surgery. For colon specifically, dementia, liver disease, other cancer and dMMR were also predictors, while obesity was found to be a predictor of receiving treatment. For rectum, nerve disease, neoadjuvant treatment and being underweight were predictors of no treatment. The adjusted OR for hospitals varies from 0.6 to 4.5 and 1.8 to 5.4 for colon and rectum, respectively. In sensitivity analyses, a low volume tertile was a predictor in colon (OR 0.77, 95% CI 0.65–0.92), but not in rectum. It should be noted that for rectum, the two hospitals with the lowest numbers had only 11 and 44 cases in the whole period, but even without these, the resulting OR of a low volume tertile was still insignificant. Some other variables were significantly associated with the outcome, but had a 95% CI very close to 1.

## 4. Discussion

### 4.1. A Review of Aim and Results

We aimed to elucidate predictors and between-hospital variations in refraining from curatively intended surgery in potentially curable colorectal cancer. We analysed this in a cohort study of patients diagnosed with CRC during the years 2009–2018, using the Danish national registers. Similarly, we aimed to examine predictors and between-hospital variations in refraining from adjuvant chemotherapy recommended in national CRC treatment guidelines. We wish to emphasize that we are confident that the actual treatment choices in each case were made for good reasons, and we did not intend to judge hospitals whose treatment patterns seemed to deviate from those of others. We only wanted to identify overall predictors for the treatment choices and to investigate whether these predictors could explain between-hospital variation in adjusted analyses.

Generally speaking, the most clinically significant results were those whose coefficients (RR or OR) deviated markedly from 1 (e.g., 0.75 or 1.25), but also had 95% confidence intervals which deviated markedly from 1 (e.g., not including 0.90 or 1.10). In light of this, we found, for the multivariable analysis of overall treatment aim, that advanced age, ASA and PS, as well as distant metastases at diagnosis and being underweight, were predictors of all non-curative treatment aims. Apart from BMI, all of these are well-established predictors for adverse outcomes. For OT-NCUR, dementia and other cancer were also predictors, compared with an aim of curatively intended surgery. Specifically for non-operative treatment, having a rectal tumor, being in the lower two quartiles of household income and living alone were predictors compared with OT-CUR. For NOT-CO, dementia, liver disease and other cancer were found to be predictors, which is not surprising. The between-hospital variations in the overall aim of treatment were found to vary significantly even after multivariable adjustments for all three non-curative aims of treatment. It may be noted that the treatment choices did not seem complementary, i.e., hospitals with a significantly positive coefficient for NOT-NO generally also had a positive coefficient for NOT-CO.

We found that refraining from adjuvant treatment was predicted by high age and ASA, kidney disease, postoperative complications and living alone, both for colon and rectum cancer. Age and ASA are markers for patient frailty, and kidney disease is important for tolerance of adjuvant chemotherapy [3]. Postoperative complications can lead to a prolonged hospital stay and deranged performance and thereby affect the presumed effect and tolerability of chemotherapy [19]. For colon cancer specifically, PS, dementia and dMMR were also predictors. For rectum cancer specifically, nerve disease, neoadjuvant treatment and being underweight were also predictors. The between-hospital variations in use of adjuvant chemotherapy were significant even after multivariable adjustments in both colon and rectum cancer.

### 4.2. A Discussion of Related Studies

As shown in Table 1, we found a distribution between treatment strategies of 90% (OT-CUR), 5% (OT-NCUR), 3% (NOT-NO) and 3% (NOT-CO). We have no knowledge of other studies that are directly comparable. Giesen et al. had a mean of 95% resected patients for non-metastatic CRC in the Netherlands [20]. We found being underweight to be a predictor of non-curative treatment. In an earlier study, we found obesity to be protective of 90-day mortality [9], which seems in line with our findings in the present study. Axt et al. found that weight loss was associated with postoperative complications [21], unfortunately we have no information on pretreatment weight loss. We also found that living alone was associated with non-operative treatment aims. It is well established that lower socioeconomic status is associated with an adverse outcome of CRC in population-based studies [7,22,23] and this correlates with our findings of low income being associated with NOT-NO. In our study, not all socioeconomic factors were predictors. As shown by others, the effect of each socioeconomic factor does not necessarily point in the same direction due to different causal mechanisms and associations with specific covariates and outcomes [24,25]. We found variations between hospitals in all three non-curative treatment aims, however with rather low numbers, especially for OT-NCUR. Giesen et al. found that resection rates vary between hospitals in the Netherlands, but that they did not translate into differences in mortality [20]. In a Swedish study, Ljunggren et al. found variations in metastatic surgery for CRC between university hospitals and non-university hospitals, but no difference in the hospital volume of patients [26], while an English study by Downing et al. found that hospitals participating in interventional studies had better survival than those who did not [27]. We did not include data on university hospitals nor participation in interventional studies. Regarding adjuvant chemotherapy treatment, Babaei et al. found between-hospital variations in stage II with risk factors (17–38%) and stage III (55–68%) in a register-based study conducted in the Netherlands, Sweden and Belgium [28]. If you ask the oncologists about adherence to national guidelines on adjuvant therapy in the Netherlands, 66% and 84% agreement was found for stage II and III, respectively [29]. Shared decision-making is on the rise, also in Denmark, but is not implemented in guidelines yet [30]. Probably this had little effect on between-hospital variations in our study of patients from the years 2009 to 2018. The goodness of fit (GOF) test of the overall analysis returned a very low *p*-value (Table 2) and the interpretation of this is important. A significant GOF test is not surprising in a large cohort and just implies to the reader that on a population level (or hospital-specific level), these relative risk ratios are plausible, but should not be used on an individual level.

We also found that 28% of the patients with an indication for adjuvant chemotherapy did not have any adjuvant treatment, with differences between stage II (58%) and III (19%). DCCG has set a desired target level for commencing adjuvant therapy of 80–90% for stage III CRC [1], which is in line with our findings, as seen from Table 3. In the latest annual report from the United Kingdom, 61% of stage III colon cancer received a major resection followed by adjuvant chemotherapy in 2019 (latest pre-COVID-19 pandemic numbers) [31]. We found that refraining from adjuvant treatment was predicted by living alone, which is in line with the socioeconomic indicators found in a recent systematic review and meta-analysis [32]. Our findings of variations between hospitals are in line with annual reports from the United Kingdom and Denmark, although the latter is with rather low numbers per hospital [1,31].

### 4.3. Strength and Limitations

This study has some limitations. First, the period these patients are collected from is rather old (2009–2018) and this can affect the generalizability of our results. Secondly, we have missing data on some variables such as reason for no referral for adjuvant therapy evaluation, WHO performance status (which is missing in the majority of patients as seen in Table 1) and clinical TN category at diagnosis (although metastases at diagnosis is included). Third, it is worth mentioning that we pooled ypT and pT categories in the analyses on adjuvant therapy. Those with a good response on neoadjuvant therapy probably have less of an effect from adjuvant therapy and vice versa [2], and this could affect the adjuvant treatment decision. We have insufficient data on this and we suspect this to have little impact on our overall conclusions. Fourth, we did not include genetic markers, such as KRAS. ESMO guidelines do not recommend the inclusion of these, due to a lack of utility in the adjuvant treatment decision-making process [3]. The strengths of this study are mainly its size, including a large cohort with almost 100% coverage of the national population [33], but also the prospective collection of data (although retrospectively extracted) from the national registries.

### 4.4. Perspectives

Refraining from curative treatment is more widespread in some hospitals than in others, even when adjusting for various predictors. No desirable proportions per hospital are applicable, since these discussions are complex with patients at a very vulnerable point in their lives and the decisions are a subject of personal preferences. Further elaboration into this research area requires further data on the impact of treatment choices on survival, morbidity and quality of life. Also, research into tools aiding shared decision-making could probably promote further equity for these patients in making well-informed decisions.

## 5. Conclusions

An intended non-curative aim of treatment in colorectal cancer was associated with age, performance status and distant metastases. Refraining from indicated adjuvant chemotherapy was also associated with age and performance status, but also with living alone, kidney disease and postoperative complications. Between-hospital variations in treatment choices were found, even in adjusted analyses, and should be examined further.

## Figures and Tables

**Figure 1 cancers-16-00366-f001:**
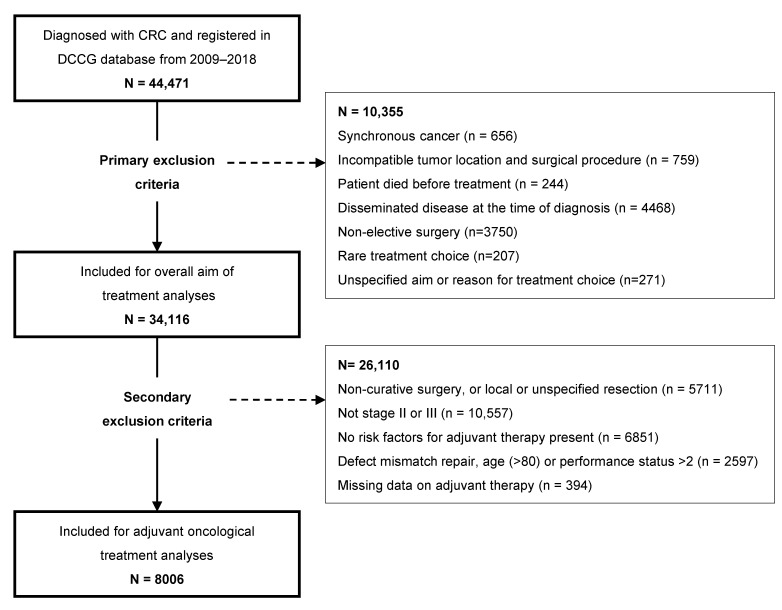
Flowchart of inclusion and exclusion in analyses. Abbreviations: CRC, colorectal cancer; DCCG, Danish Colorectal Cancer Group.

**Figure 2 cancers-16-00366-f002:**
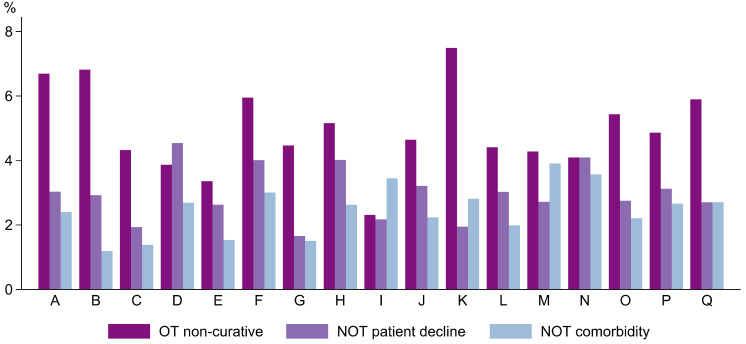
Proportions of non-curative aim of treatment by hospital (A–Q). Abbreviations: OT, operative treatment; NOT, non-operative treatment.

**Figure 3 cancers-16-00366-f003:**
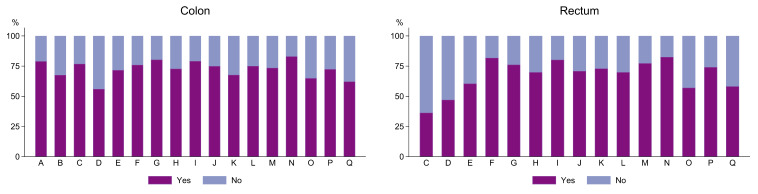
Proportion of patients receiving adjuvant chemotherapy by hospital (A–Q). Hospital A and B only treat colon cancer.

**Table 1 cancers-16-00366-t001:** Characteristics of 34,116 patients with colorectal cancer diagnosed in the years 2009–2018 in Denmark by overall treatment aim.

	OT-CUR (N = 30,548)		OT-NCUR (N = 1678)		NOT-NO (N = 1007)		NOT-CO (N = 883)		Total (N = 34,116)	
	N	%	N	%	N	%	N	%	N	%
Sex										
Male	16,777	55	897	53	498	49	497	56	18,669	55
Female	13,771	45	781	47	509	51	386	44	15,447	45
Age group										
<50	1261	4	70	4	15	1	0	0	1346	4
50–64	7612	25	304	18	70	7	32	4	8018	24
65–74	11,683	38	496	30	146	14	151	17	12,476	37
75–84	8074	26	520	31	313	31	345	39	9252	27
85+	1918	6	288	17	463	46	355	40	3024	9
ASA score										
I	7162	23	180	11	<45	4	<5	0	7387	22
II	16,793	55	781	47	<235	23	<75	8	17,882	52
III	5835	19	557	33	289	29	387	44	7068	21
IV + V	264	1	101	6	43	4	133	15	541	2
Unknown	494	2	59	4	399	40	286	32	1238	4
WHO performance status										
0	11,421	37	230	14	75	7	10	1	11,736	34
1	3782	12	246	15	122	12	74	8	4224	12
2	1054	3	160	10	134	13	159	18	1507	4
3 + 4	248	1	78	5	105	10	246	28	677	2
Unknown	14,043	46	964	57	571	57	394	45	15,972	47
Location of cancer										
Colon	20,058	66	1097	65	560	56	514	58	22,229	65
Rectum	10,490	34	581	35	447	44	369	42	11,887	35
cM category										
cM0	27,514	90	488	29	650	65	629	71	29,281	86
cM1	2686	9	1142	68	223	22	168	19	4219	12
Unknown	348	1	48	3	134	13	86	10	616	2
MDT conference										
Yes	19,678	64	1068	64	435	43	485	55	21,666	64
No	8496	28	383	23	182	18	160	18	9221	27
Unknown	2374	8	227	14	390	39	238	27	3229	9
Charlson score										
0	23,180	76	1087	65	609	60	342	39	25,218	74
1	2569	8	154	9	104	10	124	14	2951	9
2	3247	11	202	12	177	18	211	24	3837	11
3+	1552	5	235	14	117	12	206	23	2110	6
Smoking status										
Never a smoker	10,969	36	520	31	179	18	142	16	11,810	35
Ex-smoker	11,377	37	555	33	178	18	223	25	12,333	36
Active smoker	5132	17	296	18	89	9	97	11	5614	16
Unknown	3070	10	307	18	561	56	421	48	4359	13
Alcohol consumed per week (units ^1^)										
0–14	24,024	79	1249	74	415	41	412	47	26,100	77
>14	3745	12	147	9	46	5	53	6	3991	12
Unknown	2779	9	282	17	546	54	418	47	4025	12
WHO body mass index class										
Underweight	741	2	111	7	38	4	57	6	947	3
Normal	12,014	39	738	44	279	28	258	29	13,289	39
Overweight	10,878	36	428	26	164	16	143	16	11,613	34
Obese	5231	17	206	12	61	6	54	6	5552	16
Unknown	1684	6	195	12	465	46	371	42	2715	8
Highest educational level ^2^										
Short	10,684	35	728	43	461	46	429	49	12,302	36
Medium	13,635	45	620	37	302	30	299	34	14,856	44
Long	5363	18	227	14	96	10	86	10	5772	17
Unknown or unclassified	866	3	103	6	148	15	69	8	1186	3
Annual household income										
1st quartile	6909	23	557	33	404	40	311	35	8181	24
2nd quartile	7257	24	451	27	332	33	332	38	8372	25
3rd quartile	7857	26	385	23	172	17	154	17	8568	25
4th quartile	8419	28	276	16	94	9	86	10	8875	26
Unknown	106	0	9	0	5	0	0	0	120	0
Cohabitation status										
Cohabiting	19,096	63	873	52	311	31	303	34	20,583	60
Alone	<11,415	37	<805	48	<700	69	580	66	13,492	40
Unknown	<40	0	<5	0	<5	0	0	0	41	0

In order to avoid showing identifiable data some numbers have been changed to <N. ^1^ One unit = 12 g of pure ethanol. ^2^ International Standard Classification of Education (ISCED) 2011. Abbreviations: ASA, American Society of Anesthesiologists; WHO, World Health Organization; MDT, treatment decisive multidisciplinary team conference; OT-CUR, operative treatment with curative intent; OT-NCUR, operative treatment with compromised/palliative intent; NOT-NO, non-operative treatment due to patient decline; NOT-CO, non-operative treatment due to patient comorbidity.

**Table 2 cancers-16-00366-t002:** Multivariable multinomial logistic regression of a non-curative aim of treatment with curative surgical treatment as the comparison level for 34,116 colorectal cancer patients.

	OT-NCUR			NOT-NO			NOT-CO		
	RR	95% CI	*p*	RR	95% CI	*p*	RR	95% CI	*p*
Sex (ref. female)									
Male	1.1	[0.99–1.27]	0.072	1.2	[1.03–1.45]	**0.022**	1.4	[1.20–1.75]	**0.000**
Age group (ref. 50–64)									
<50	1.1	[0.81–1.51]	0.520	1.2	[0.66–2.35]	0.495	-	-	-
65–74	1.0	[0.84–1.19]	0.985	1.2	[0.89–1.70]	0.210	2.4	[1.55–3.73]	**0.000**
75–84	1.3	[1.09–1.58]	**0.004**	2.5	[1.85–3.49]	**0.000**	4.6	[2.99–7.15]	**0.000**
85+	3.4	[2.73–4.34]	**0.000**	10.8	[7.74–15.13]	**0.000**	16.6	[10.51–26.09]	**0.000**
ASA score (ref. I)									
II	1.5	[1.23–1.82]	**0.000**	1.0	[0.70–1.45]	0.983	2.7	[0.85–8.86]	0.090
III	2.1	[1.69–2.67]	**0.000**	1.3	[0.90–1.98]	0.156	10.0	[3.11–32.00]	**0.000**
IV + V	7.4	[5.13–10.63]	**0.000**	2.9	[1.72–4.98]	**0.000**	41.9	[12.70–138.17]	**0.000**
Unknown	2.5	[1.71–3.62]	**0.000**	15.3	[10.02–23.37]	**0.000**	100.1	[30.82–324.87]	**0.000**
WHO performance status (ref. 0)									
1	2.2	[1.74–2.67]	**0.000**	2.9	[2.10–3.99]	**0.000**	7.6	[3.85–15.20]	**0.000**
2	4.4	[3.39–5.75]	**0.000**	8.3	[5.88–11.68]	**0.000**	33.8	[17.12–66.55]	**0.000**
3 + 4	7.1	[4.97–10.19]	**0.000**	21.4	[14.34–31.83]	**0.000**	146.2	[73.10–292.60]	**0.000**
Unknown	1.5	[1.13–1.88]	**0.003**	1.7	[1.16–2.43]	**0.006**	6.4	[3.17–12.95]	**0.000**
Location of cancer (ref. colon)									
Rectum	1.2	[1.06–1.39]	**0.004**	3.4	[2.86–4.05]	**0.000**	3.2	[2.67–3.92]	**0.000**
cM category (ref. cM0)									
cM1	28.9	[25.47–32.81]	**0.000**	4.6	[3.76–5.61]	**0.000**	4.1	[3.23–5.11]	**0.000**
Unknown	5.1	[3.65–7.20]	**0.000**	4.3	[3.05–6.01]	**0.000**	3.2	[2.22–4.69]	**0.000**
MDT conference (ref. yes)									
No	0.9	[0.77–1.06]	0.231	1.1	[0.85–1.36]	0.558	0.7	[0.57–0.96]	**0.021**
Unknown	0.9	[0.71–1.15]	0.424	4.0	[2.88–5.51]	**0.000**	2.2	[1.52–3.16]	**0.000**
Comorbidity									
Cardiovascular disease	0.9	[0.78–1.02]	0.091	1.1	[0.88–1.32]	0.454	1.4	[1.06–1.75]	**0.015**
Chronic pulmonary disease	1.0	[0.85–1.11]	0.674	0.9	[0.78–1.11]	0.412	1.2	[0.99–1.43]	0.061
Diabetes	0.9	[0.79–1.12]	0.508	1.0	[0.80–1.24]	0.965	1.1	[0.90–1.39]	0.299
Dementia	2.2	[1.55–3.10]	**0.000**	1.2	[0.78–1.71]	0.471	1.8	[1.27–2.58]	**0.001**
Liver disease	1.3	[0.75–2.31]	0.342	1.1	[0.51–2.35]	0.812	4.3	[2.52–7.19]	**0.000**
Kidney disease	1.1	[0.75–1.50]	0.744	1.2	[0.83–1.72]	0.342	1.4	[1.02–1.94]	**0.040**
Nerve disease	1.2	[0.72–1.90]	0.522	0.7	[0.39–1.34]	0.303	0.9	[0.54–1.57]	0.776
Other cancer	1.4	[1.18–1.61]	**0.000**	1.1	[0.86–1.37]	0.473	1.4	[1.10–1.75]	**0.006**
Connective tissue disease	1.0	[0.78–1.18]	0.689	0.9	[0.70–1.18]	0.455	1.1	[0.86–1.42]	0.453
Affective disorder	0.9	[0.78–1.04]	0.163	1.1	[0.92–1.33]	0.274	1.3	[1.07–1.56]	**0.008**
Schizophrenia spectrum disorder	0.8	[0.36–1.65]	0.503	2.3	[1.06–4.90]	**0.035**	2.3	[1.01–5.31]	**0.048**
Personality and behavior disorder	0.6	[0.20–1.95]	0.413	1.9	[0.55–6.34]	0.317	1.2	[0.27–5.63]	0.787
Psychoactive drug abuse disorder	0.9	[0.70–1.26]	0.681	0.9	[0.64–1.40]	0.782	1.2	[0.80–1.66]	0.445
Smoking status (ref. non-smoker)									
Ex-smoker	1.0	[0.86–1.14]	0.865	0.9	[0.68–1.08]	0.195	1.0	[0.80–1.33]	0.823
Active smoker	1.0	[0.86–1.22]	0.788	1.2	[0.87–1.55]	0.323	1.1	[0.83–1.59]	0.413
Unknown	1.2	[0.95–1.61]	0.115	2.0	[1.38–2.78]	**0.000**	1.4	[0.97–2.12]	0.074
Alcohol consumed per week (units ^1^) (ref. 0–14)									
>14	0.8	[0.65–0.99]	**0.037**	1.1	[0.77–1.51]	0.665	1.1	[0.75–1.52]	0.734
Unknown	1.1	[0.85–1.43]	0.475	1.8	[1.31–2.56]	**0.000**	1.8	[1.26–2.60]	**0.001**
WHO Body mass index class (ref. normal weight)									
Underweight	1.9	[1.47–2.48]	**0.000**	1.6	[1.11–2.44]	**0.013**	2.3	[1.58–3.39]	**0.000**
Overweight	0.7	[0.65–0.86]	**0.000**	0.9	[0.69–1.05]	0.139	0.8	[0.61–1.00]	**0.046**
Obese	0.8	[0.67–0.97]	**0.022**	0.7	[0.53–0.98]	**0.037**	0.6	[0.39–0.78]	**0.001**
Unknown	0.9	[0.73–1.21]	0.620	1.7	[1.25–2.21]	**0.001**	2.5	[1.88–3.42]	**0.000**
Highest educational level ^2^ (ref. long)									
Short	1.1	[0.90–1.32]	0.383	0.9	[0.68–1.20]	0.467	0.9	[0.69–1.31]	0.747
Medium	1.0	[0.81–1.17]	0.753	0.8	[0.60–1.06]	0.120	0.9	[0.63–1.18]	0.356
Unknown or unclassified	1.3	[0.97–1.81]	0.073	1.4	[0.93–1.98]	0.109	0.9	[0.58–1.42]	0.681
Annual household income (ref. 4th quartile)									
1st quartile	1.2	[0.97–1.45]	0.102	1.5	[1.10–2.03]	**0.011**	1.4	[0.96–1.92]	0.086
2nd quartile	1.2	[0.97–1.44]	0.089	1.6	[1.20–2.16]	**0.002**	1.4	[1.03–2.00]	**0.031**
3rd quartile	1.1	[0.95–1.37]	0.166	1.2	[0.92–1.68]	0.148	1.0	[0.71–1.41]	0.994
Unknown	1.8	[0.67–4.58]	0.250	3.0	[0.72–12.78]	0.130	-	-	-
Cohabitation status (ref. cohabiting)									
Alone	1.2	[1.03–1.32]	**0.017**	1.7	[1.39–1.97]	**0.000**	1.4	[1.14–1.66]	**0.001**
Unknown	0.8	[0.13–4.90]	0.803	0.3	[0.01–4.52]	0.349	-	-	-
Hospital (ref. Q)									
A	1.2	[0.85–1.80]	0.272	1.8	[1.04–3.26]	**0.037**	1.2	[0.63–2.15]	0.635
B	1.1	[0.76–1.62]	0.581	0.6	[0.33–0.99]	**0.044**	0.2	[0.12–0.49]	**0.000**
C	0.6	[0.42–0.90]	**0.012**	0.6	[0.31–1.03]	0.064	0.4	[0.20–0.73]	**0.004**
D	0.7	[0.50–1.05]	0.092	1.8	[1.16–2.77]	**0.009**	1.0	[0.57–1.58]	0.857
E	0.3	[0.23–0.48]	**0.000**	0.9	[0.57–1.48]	0.716	0.4	[0.22–0.73]	**0.003**
F	1.2	[0.90–1.62]	0.216	2.6	[1.74–3.92]	**0.000**	1.8	[1.13–2.74]	**0.012**
G	0.9	[0.70–1.28]	0.709	0.7	[0.43–1.14]	0.151	0.5	[0.29–0.81]	**0.006**
H	1.0	[0.73–1.30]	0.870	2.8	[1.93–4.17]	**0.000**	1.5	[0.95–2.27]	0.085
I	0.3	[0.24–0.49]	**0.000**	1.1	[0.73–1.77]	0.583	1.7	[1.07–2.56]	**0.023**
J	0.8	[0.57–1.01]	0.060	1.8	[1.17–2.65]	**0.007**	1.5	[0.94–2.37]	0.087
K	1.1	[0.83–1.37]	0.613	0.8	[0.52–1.22]	0.300	1.2	[0.79–1.81]	0.399
L	0.6	[0.44–0.78]	**0.000**	1.3	[0.85–1.87]	0.254	0.7	[0.44–1.09]	0.115
M	0.7	[0.50–0.89]	**0.005**	1.0	[0.68–1.52]	0.923	1.4	[0.92–2.02]	0.128
N	0.8	[0.58–1.02]	0.068	2.0	[1.40–2.87]	**0.000**	1.5	[1.04–2.28]	**0.030**
O	0.8	[0.58–0.99]	**0.040**	0.8	[0.57–1.25]	0.401	0.6	[0.38–0.89]	**0.013**
P	1.3	[0.96–1.64]	0.100	2.6	[1.75–3.74]	**0.000**	2.1	[1.41–3.20]	**0.000**
Year of diagnosis	0.9	[0.88–0.96]	**0.000**	1.2	[1.14–1.27]	**0.000**	1.2	[1.12–1.25]	**0.000**
Intercept	0.0	[0.01–0.01]	**0.000**	0.0	[0.00–0.00]	**0.000**	0.0	[0.00–0.00]	**0.000**

^1^ One unit = 12 g of pure ethanol. ^2^ International Standard Classification of Education (ISCED) 2011. *p*-values < 0.05 in bold. Abbreviations: RR, relative risk ratio; CI, confidence interval; ASA, American Society of Anesthesiologists; WHO, World Health Organization; MDT, treatment decisive multidisciplinary team; OT-CUR, operative treatment with curative intent; OT-NCUR, operative treatment with compromised/palliative intent; NOT-NO, non-operative treatment due to patient decline; NOT-CO, non-operative treatment due to patient comorbidity. Goodness of fit test: *p*-value= 0.000.

**Table 3 cancers-16-00366-t003:** Overview of 8006 colorectal cancer patients with an indication for adjuvant chemotherapy in the years 2009–2018.

	Colon						Rectum					
	Treatment with Adjuvant Therapy						Treatment with Adjuvant Therapy					
	No treatment		Treatment		Total		No treatment		Treatment		Total	
	N	%	N	%	N	%	N	%	N	%	N	%
Sex												
Male	818	55	2151	53	2969	53	456	62	1055	61	1511	62
Female	682	45	1909	47	2591	47	274	38	661	39	935	38
Age group												
<50	15	1	251	6	266	5	24	3	143	8	167	7
50–64	289	19	1348	33	1637	29	184	25	702	41	886	36
65–74	679	45	1881	46	2560	46	336	46	691	40	1027	42
75–79	517	34	580	14	1097	20	186	25	180	10	366	15
ASA score												
I	205	14	1243	31	1448	26	133	18	636	37	769	31
II	758	51	2323	57	3081	55	423	58	912	53	1335	55
III	475	32	443	11	918	17	167	23	147	9	314	13
IV + V	36	2	7	0	43	1	<10	1	<5	0	8	0
Unknown	26	2	44	1	70	1	<5	0	<20	1	20	1
WHO performance status												
0	374	25	1601	39	1975	36	175	24	751	44	926	38
1	216	14	380	9	596	11	72	10	99	6	171	7
2	80	5	63	2	143	3	<25	3	<15	1	34	1
3 + 4	26	2	8	0	34	1	<5	0	<5	0	5	0
Unknown	804	54	2008	49	2812	51	457	63	853	50	1310	54
UICC stage												
II	695	46	557	14	1252	23	329	45	174	10	503	21
III	805	54	3503	86	4308	77	401	55	1542	90	1943	79
Neoadjuvant therapy												
No							502	69	1468	86	1970	81
Yes							228	31	248	14	476	19
Mismatch repair status												
pMMR	971	65	2918	72	3889	70	517	71	1381	80	1898	78
dMMR	251	17	422	10	673	12	15	2	22	1	37	2
Missing MMR	278	19	720	18	998	18	198	27	313	18	511	21
Postoperative medical complication < 30 days												
No	1136	76	3559	88	4695	84	531	73	1565	91	2096	86
Yes	237	16	167	4	404	7	128	18	83	5	211	9
Unknown	127	8	334	8	461	8	71	10	68	4	139	6
Postoperative surgical complication < 30 days												
No	1018	68	3271	81	4289	77	375	51	1253	73	1628	67
Yes	360	24	468	12	828	15	290	40	396	23	686	28
Unknown	122	8	321	8	443	8	65	9	67	4	132	5
Charlson score												
0	1008	67	3341	82	4349	78	543	74	1476	86	2019	83
1	160	11	287	7	447	8	64	9	87	5	151	6
2	223	15	317	8	540	10	89	12	121	7	210	9
3+	109	7	115	3	224	4	34	5	32	2	66	3
Smoking status												
Never a smoker	457	30	1587	39	2044	37	205	28	690	40	895	37
Ex-smoker	572	38	1493	37	2065	37	286	39	612	36	898	37
Active smoker	313	21	661	16	974	18	170	23	297	17	467	19
Unknown	158	11	319	8	477	9	69	9	117	7	186	8
Alcohol consumed per week (units ^1^)												
0–14	1181	79	3269	81	4450	80	559	77	1394	81	1953	80
>14	185	12	511	13	696	13	100	14	221	13	321	13
Unknown	134	9	280	7	414	7	71	10	101	6	172	7
WHO Body mass index class												
Underweight	47	3	73	2	120	2	35	5	17	1	52	2
Normal	582	39	1513	37	2095	38	284	39	659	38	943	39
Overweight	502	33	1471	36	1973	35	254	35	671	39	925	38
Obese	269	18	821	20	1090	20	124	17	316	18	440	18
Unknown	100	7	182	4	282	5	33	5	53	3	86	4
Highest educational level ^2^												
Short	600	40	1216	30	1816	33	306	42	476	28	782	32
Medium	655	44	1923	47	2578	46	301	41	835	49	1136	46
Long	203	14	836	21	1039	19	104	14	373	22	477	20
Unknown or unclassified	42	3	85	2	127	2	19	3	32	2	51	2
Annual household income												
1st quartile	425	28	704	17	1129	20	211	29	266	16	477	20
2nd quartile	411	27	848	21	1259	23	160	22	320	19	480	20
3rd quartile	375	25	1127	28	1502	27	194	27	475	28	669	27
4th quartile	283	19	1374	34	1657	30	160	22	650	38	810	33
Unknown	6	0	7	0	13	0	5	1	5	0	10	0
Cohabitation status												
Cohabiting	873	58	2879	71	3752	67	444	61	1256	73	1700	70
Alone	<630	42	<1185	29	<1810	32	<285	39	<460	27	<745	30
Unknown	<5	0	<5	0	<5	0	<5	0	<5	0	<5	0

In order to avoid showing identifiable data some numbers are changed to <n. ^1^ One unit = 12 g of pure ethanol. ^2^ International Standard Classification of Education (ISCED) 2011. Abbreviations: ASA, American Society of Anesthesiologists; WHO, World Health Organization; MDT, treatment decisive multidisciplinary team; pMMR, proficient mismatch repair; dMMR, deficient mismatch repair.

**Table 4 cancers-16-00366-t004:** Multivariable logistic regression of receiving adjuvant chemotherapy for 8006 stage II or III colorectal cancer patients with an indication for adjuvant chemotherapy treatment.

	Colon			Rectum		
	OR	95% CI	*p*	OR	95%CI	*p*
Sex (ref. female)						
Male	0.90	[0.77–1.04]	0.161	1.18	[0.93–1.48]	0.170
Age group (ref. 50–64)						
<50	3.68	[2.09–6.51]	**0.000**	1.44	[0.84–2.47]	0.180
65–74	0.79	[0.65–0.94]	**0.010**	0.62	[0.47–0.80]	**0.000**
75–79	0.34	[0.27–0.42]	**0.000**	0.28	[0.20–0.38]	**0.000**
N-a	0.93	[0.54–1.60]	0.790	0.23	[0.09–0.54]	**0.001**
ASA score (ref. I)						
II	0.78	[0.63–0.95]	**0.016**	0.77	[0.58–1.02]	0.066
III	0.35	[0.27–0.46]	**0.000**	0.48	[0.32–0.72]	**0.000**
IV + V	0.09	[0.04–0.22]	**0.000**	0.58	[0.09–3.70]	0.565
Unknown	0.45	[0.24–0.84]	**0.011**	3.79	[0.75–19.32]	0.108
WHO performance status (ref. 0)						
1	0.81	[0.64–1.04]	0.096	0.63	[0.41–0.95]	**0.029**
2	0.63	[0.41–0.97]	**0.037**	0.43	[0.18–1.04]	0.062
3 + 4	0.29	[0.11–0.72]	**0.008**	0.22	[0.02–2.51]	0.224
N-a ^1^						
Unknown	1.32	[0.99–1.76]	0.055	1.28	[0.82–2.02]	0.280
Comorbidity						
Cardiovascular disease	0.93	[0.79–1.10]	0.407	0.90	[0.71–1.15]	0.402
Chronic pulmonary disease	0.96	[0.81–1.13]	0.607	0.99	[0.76–1.30]	0.947
Diabetes	0.80	[0.65–0.97]	**0.027**	0.82	[0.60–1.13]	0.234
Dementia	0.28	[0.12–0.66]	**0.004**	0.37	[0.12–1.15]	0.085
Liver disease	0.48	[0.25–0.91]	**0.024**	0.59	[0.14–2.49]	0.476
Kidney disease	0.41	[0.25–0.66]	**0.000**	0.29	[0.12–0.74]	**0.009**
Nerve disease	0.62	[0.32–1.22]	0.166	0.25	[0.08–0.81]	**0.020**
Other cancer	0.74	[0.59–0.93]	**0.010**	0.77	[0.53–1.12]	0.175
Connective tissue disease	0.77	[0.59–1.01]	0.055	0.88	[0.57–1.34]	0.539
Affective disorder	0.87	[0.73–1.04]	0.124	1.04	[0.78–1.39]	0.805
Schizophrenia spectrum disorder	0.36	[0.13–1.01]	0.052	0.36	[0.07–1.88]	0.227
Disorder of adult personality and behaviour	1.02	[0.29–3.59]	0.981	0.25	[0.06–1.04]	0.056
Psychoactive drug abuse disorder	0.98	[0.71–1.36]	0.906	1.00	[0.59–1.70]	0.993
Microsatellite status of tumor (ref. pMMR)						
dMMR	0.41	[0.33–0.51]	**0.000**	0.62	[0.22–1.73]	0.357
N-a	0.29	[0.25–0.35]	**0.000**	1.22	[0.78–1.93]	0.384
Missing MMR	0.77	[0.58–1.00]	0.051	1.01	[0.63–1.61]	0.968
Postoperative medical complication < 30 days (ref. no)						
Yes	0.41	[0.32–0.52]	**0.000**	0.28	[0.20–0.40]	**0.000**
Unknown	1.14	[0.67–1.93]	0.624	0.59	[0.25–1.42]	0.237
Postoperative surgical complication < 30 days (ref. no)						
Yes	0.46	[0.38–0.56]	**0.000**	0.41	[0.33–0.52]	**0.000**
Unknown	0.89	[0.53–1.51]	0.669	1.21	[0.49–2.96]	0.680
Smoking status (ref. non-smoker)						
Ex-smoker	0.96	[0.82–1.14]	0.675	0.76	[0.59–0.97]	**0.031**
Active smoker	0.84	[0.68–1.04]	0.112	0.74	[0.54–1.00]	**0.048**
Unknown	1.01	[0.69–1.47]	0.964	1.19	[0.69–2.05]	0.536
Alcohol consumed per week (units ^2^) (ref. 0–14)						
>14	1.05	[0.84–1.31]	0.682	0.93	[0.68–1.27]	0.638
Unknown	1.11	[0.74–1.65]	0.616	0.52	[0.31–0.90]	**0.019**
WHO body mass index class (ref. normal weight)						
Underweight	0.78	[0.49–1.23]	0.289	0.30	[0.15–0.62]	**0.001**
Overweight	1.12	[0.95–1.32]	0.191	1.29	[1.01–1.65]	**0.038**
Obese	1.46	[1.19–1.80]	**0.000**	1.35	[0.99–1.84]	0.056
Unknown	0.90	[0.61–1.32]	0.591	1.34	[0.68–2.64]	0.403
Highest educational level ^3^ (ref. long)						
Short	0.77	[0.62–0.97]	**0.026**	0.79	[0.57–1.10]	0.167
Medium	0.84	[0.68–1.04]	0.106	0.97	[0.72–1.32]	0.867
Unknown or unclassified	0.67	[0.42–1.06]	0.088	0.98	[0.45–2.14]	0.951
Annual household income (ref. 4th quartile)						
1st quartile	0.78	[0.62–0.99]	**0.042**	0.86	[0.61–1.22]	0.406
2nd quartile	0.87	[0.69–1.08]	0.199	1.02	[0.73–1.42]	0.927
3rd quartile	0.88	[0.72–1.08]	0.207	0.95	[0.71–1.27]	0.721
Unknown	0.20	[0.04–0.92]	**0.039**	0.11	[0.02–0.72]	**0.021**
Cohabitation status (ref. cohabiting)						
Alone	0.70	[0.60–0.82]	**0.000**	0.70	[0.56–0.89]	**0.003**
Unknown	0.29	[0.01–12.22]	0.517	1.83	[0.10–32.09]	0.680
Year of surgery	1.13	[1.07–1.20]	**0.000**	1.17	[1.02–1.33]	**0.021**
Hospital (ref. Q)						
A	3.57	[2.29–5.57]	**0.000**	-	-	-
B	1.24	[0.83–1.83]	0.295	-	-	-
C	2.56	[1.69–3.89]	**0.000**	0.77	[0.18–3.29]	0.722
D	0.64	[0.43–0.96]	**0.030**	0.55	[0.29–1.06]	0.073
E	1.59	[1.06–2.40]	**0.026**	1.38	[0.73–2.61]	0.317
F	2.50	[1.74–3.58]	**0.000**	5.53	[1.98–15.41]	**0.001**
G	3.36	[2.25–5.01]	**0.000**	2.45	[1.45–4.14]	**0.001**
H	1.82	[1.26–2.62]	**0.001**	1.82	[1.04–3.20]	**0.036**
I	2.97	[2.01–4.38]	**0.000**	2.77	[1.61–4.77]	**0.000**
J	2.21	[1.55–3.16]	**0.000**	1.91	[1.16–3.15]	**0.011**
K	1.96	[1.28–3.02]	**0.002**	2.20	[1.42–3.40]	**0.000**
L	2.76	[1.93–3.97]	**0.000**	2.21	[1.37–3.58]	**0.001**
M	2.35	[1.65–3.35]	**0.000**	2.49	[1.43–4.33]	**0.001**
N	4.42	[3.08–6.37]	**0.000**	4.02	[2.42–6.68]	**0.000**
O	1.37	[0.97–1.93]	0.072	1.12	[0.73–1.72]	0.607
P	2.19	[1.58–3.02]	**0.000**	2.50	[1.51–4.14]	**0.000**
Neoadjuvant therapy (ref. no)						
Yes	-	-	-	0.32	[0.24–0.43]	**0.000**
N-a	-	-	-	0.62	[0.38–1.00]	0.051
Intercept	4.60	[2.59–8.16]	**0.000**	3.02	[1.28–7.11]	**0.011**

^1^ omitted from analysis due to collinearity with age group n-a. ^2^ One unit = 12 g of pure ethanol. ^3^ International Standard Classification of Education (ISCED) 2011. *p*-values < 0.05 in bold. Abbreviations: OR, odds ratio; CI, confidence interval; ASA, American Society of Anesthesiologists; WHO, World Health Organization; MDT, treatment decisive multidisciplinary team; pMMR, proficient mismatch repair; dMMR, deficient mismatch repair. Goodness of fit test: *p*-value 0.55 (Colon) and 0.67 (Rectum).

## Data Availability

Data were obtained under a license granted specifically for this study and cannot be made available according to Danish legislation. Researchers can apply for data at www.dst.dk, www.sundhedsdatastyrelsen.dk and www.rkkp.dk (accessed on 8 January 2024).

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
