# Peer review of "Non-Curative Treatment Choices in Colorectal Cancer: Predictors and Between-Hospital Variations in Denmark: A Population-Based Register Study"

_cancers, 2024, doi:10.3390/cancers16020366_

Round 1

Reviewer 1 Report

Comments and Suggestions for Authors

Variations in treatment choices have been reported in colorectal cancer (CRC). In this study, the authors aimed to elucidate predictors and between hospital variations in refraining from curatively intended surgery and adjuvant chemotherapy in potentially curable colorectal cancer. They found that non-curative overall treatment and refraining from adjuvant chemotherapy was associated with well-known risk factors, but the former also underweight and the latter also living alone. Marked between-hospital variations were found and should be examined further.

1. It’s suggested to follow the survival time of patients. The survival time should be used for the related analysis.

2. It’s just a retrospective study. This is one of the limitations of this study.

3. The adjuvant chemotherapy for CRC is complicated. It’s suggested to add the details related the adjuvant chemotherapy in this study.

4. In the introduction part, it’s suggested to introduce the adjuvant chemotherapy for CRC.

5. Whether is the gene mutation information (such as KRAS) of these patients related to the results?

Comments on the Quality of English Language

It's suggested to check the typo errors and correct grammar mistakes.

Author Response

We would like to thank the reviewer for thorough revision of our manuscript and we are happy to provide a revised manuscript together with a point by point cover letter to address the details of the revisions and our response. Our comments are seen below (in red). Changes are highlighted in the manuscript (in yellow)

  1. It’s suggested to follow the survival time of patients. The survival time should be used for the related analysis.
    • We acknowledge that survival is a very relevant parameter in this patient group. However, the scope of this study was to elucidate predictors and between-hospital variations in refraining from curatively intended surgery and adjuvant chemotherapy. We did not aim to examine survival differences between the groups, but this could be an aim of further studies.
  2. It’s just a retrospective study. This is one of the limitations of this study.
    • We thank the reviewer for pointing this out. This have now been added to the bottom of section 4.3.
  3. The adjuvant chemotherapy for CRC is complicated. It’s suggested to add the details related the adjuvant chemotherapy in this study.
    • Thank you for this suggestion. A small section in the methods section 2.4 has been added to explain the adjuvant regimes in the different periods.
  4. In the introduction part, it’s suggested to introduce the adjuvant chemotherapy for CRC.
    • Thank you for pointing out this flaw. A section in the background concerning the present adjuvant chemotherapy regime has now been added.
  5. Whether is the gene mutation information (such as KRAS) of these patients related to the results?
    • We thank the reviewer for this comment. We did not examine genetic markers such as KRAS and BRAF. In the ESMO guidelines on colon cancer(Argiles et al.) it is recommended not to include these in the assessment of risk of recurrence in non-metastatic cancer, based on their lack of utility in the adjuvant decision-making process (Niedzwiecki et al. 2016. Association between results of a gene expression signature assay and recurrence-free interval in patients with stage II colon cancer in cancer and leukemia group B 9581 (Alliance). doi: 10.1200/JCO.2015.65.4699).
    • We have now added a paragraph about this in section 4.3.
  6. It's suggested to check the typo errors and correct grammar mistakes.
    • We thank the reviewer for noticing this. We have now corrected several grammar mistakes throughout the manuscript.

We hope that you find the abovementioned comments and corrections to the article satisfying.

Reviewer 2 Report

Comments and Suggestions for Authors

Rattenborg S et al. have made an interesting study to highlight a subset of patients with advanced colorectal cancer were adopting non-curative treatment and refraining from adjuvant therapy. Actually, surgical and medical oncologists have been frequently encountering such patients and always managed based on the multi-disciplinary therapeutic (MDT) approach. Therefore, although such issue is interesting but is not of great clinical importance in the era of precision medicine. Therefore, I would like to point out the originality of this study is limited. However, overall, the manuscript was well written and I have no major criticisms. Furthermore, in order to enhance the comprehensiveness of this article, I would like to suggest the authors cite the similar studies from the Asian fellow researchers: 

1. Asian Journal of Surgery. Volume 46, Issue 9, September 2023, Pages 3710-3715

2. Asian Journal of Surgery. Volume 45, Issue 1, January 2022, Pages 97-104

3. Asian Journal of Surgery. Available online 1 December 2023. Impact of positron emission tomography on the surgical treatment of locoregionally recurrent colorectal cancer

4. Asian Journal of Surgery. Volume 45, Issue 1, January 2022, Pages 208-212

5. Asian Journal of Surgery. Available online 28 August 2023. Is cytoreductive surgery and hyperthermic intraperitoneal chemotherapy still beneficial in patients diagnosed with colorectal peritoneal metastasis who underwent palliative chemotherapy?

6. Asian Journal of Surgery. Volume 46, Issue 10, October 2023, Pages 4378-4384

Author Response

We would like to thank the reviewer for thorough revision of our manuscript and we are happy to provide a revised manuscript together with a point by point cover letter to address the details of the revisions and our response. Our comments are seen below (in red).

”….suggest the authors cite the similar studies from the Asian fellow researchers:

  1. Asian Journal of Surgery. Volume 46, Issue 9, September 2023, Pages 3710-3715
  2. Asian Journal of Surgery. Volume 45, Issue 1, January 2022, Pages 97-104
  3. Asian Journal of Surgery. Available online 1 December 2023. Impact of positron emission tomography on the surgical treatment of locoregionally recurrent colorectal cancer
  4. Asian Journal of Surgery. Volume 45, Issue 1, January 2022, Pages 208-212
  5. Asian Journal of Surgery. Available online 28 August 2023. Is cytoreductive surgery and hyperthermic intraperitoneal chemotherapy still beneficial in patients diagnosed with colorectal peritoneal metastasis who underwent palliative chemotherapy?
  6. Asian Journal of Surgery. Volume 46, Issue 10, October 2023, Pages 4378-4384”

We thank the reviewer for providing some very interesting literature, which encompasses a broad range of subjects. We have reviewed all the proposed studies thoroughly and found that their scopes and aims differ more or less markedly from the scope of our study (see details below). On a more general level, we have focused on discussing previous studies from countries with healthcare systems, patient-related risk factors and living conditions more similar to Denmark and the other Nordic countries. Therefore, with all due respect, we would prefer not to include these studies as references, unless Editor disagrees.

  1. The article by Ho et al from 2023 (1st suggestion by the reviewer) is a study of unresectable stage IV CRC patients. As seen in our study (Table 1) only 12% of this population had metastatic disease, since we excluded patients deemed to have “disseminated disease”. An earlier study (Clin Epidemiol. 2023 Jul 22;15:867-880) found that 24% had metastatic disease. Therefore the mentioned study does not seem to be relevant within the scope of the current study.
  2. The article by Han et al from 2022 (2nd suggestion) is also an interesting paper on those patients who had a complete clinical response on neoadjuvant chemotherapy. However these patients was not formally categorized in our cohort, due to databreaks, and therefore it is not possible for us to engage into this very interesting patient population.
  3. The article by Liang et al from 2023 (3rd suggestion) is about recurrent cancer, which is not within our scope, where we examine those with a first time diagnosis of colorectal cancer.
  4. The article by Hashimoto et al 2022 (4th suggestion) is about survival in nonagerians after elective curative treatment of colorectal cancer. This is a small retrospective study of elderly patients, including 50 elective surgically treated patients, which finds that zero patients died within 30 days after surgery. The authors conclude that performance score were significantly associated with overall survival, but with only 50 patients it seems underpowered and no comparison with non-surgically treated patients. Also it looks at survival and not treatment choices.
  5. The article by Cho et al 2023 (5th suggestion) is about cytoreductive surgery and HIPEC treatment in patients already treated with palliative chemotherapy. This does not match with the aim of our study, where we focus on those with adjuvant chemotherapy and not palliative chemotherapy.
  6. The final article by Sert et al 2023 (6th suggestion) has the aim of examining a neoadjuvant rectal cancer (NAR) score as a predictive factor for treatment outcomes after neoadjuvant chemoradiotherapy. Since we focus on treatment choices and Sert et al focus on treatment outcomes (overall survival, local recurrence-free survival and distant metastasis-free survival) the aims (as well as the eligibility criteria) deviates too much to allow for a meaningful perspectivation.

We hope that this reply is satisfying for the reviewer.

Reviewer 3 Report

Comments and Suggestions for Authors

General comments:

This is a large-scale population study using the Danish national registers that aimed to evaluate predictors and between-hospital variations in refraining from curatively intended surgery and adjuvant chemotherapy in potentially curable colorectal cancer (CRC).

The authors found that the identified predictors were mostly well-known risk factors (such as high age, poor performance, distant metastases, kidney disease, and postoperative complications). However, they also have elucidated some unique predictors including underweight for non-curative treatment and living alone for no adjuvant treatment. These results may reflect the real-world trend involving CRC patients with environmental, socioeconomic, psychiatric and medical conditions. The authors’ perspective included critical implications that between-hospital variations may be inevitable because the discussions are complex and subjected to personal preferences.

The discussion is well structured and written with the relevant reflections. The methodology Although some limitations present, this study has a significant strength in the comprehensive analysis using a nation-wide cohort with prospective data collection.

Minor points:

1.      The p-values may preferably be presented with the actual values: Although some data are “statistically significant with P<0.05” in Table 2 and 4, the actual values are very helpful for readers to interpret how significant they are.

2.      The meaning of “asterisks” in Table 4 should be displayed in the footnote.

3.      The “RRR” indicates “relative risk ratio” in this study, however it may be misunderstood as “relative risk reduction (1-RR)”. Simply, “RR” may be more suitable.

4.      During the study period (2009-2018), what could be the major or optional regimens for neo- and adjuvant therapy in this cohort?

5.      Further studies are warranted to examine the oncologic impact of the observed predictors on patients’ survival. Please propose the authors’ possible future investigations.

Author Response

We would like to thank the reviewer for thorough revision of our manuscript and we are happy to provide a revised manuscript together with a point by point cover letter to address the details of the revisions and our response. Our comments are seen below (in red).

  1. The p-values may preferably be presented with the actual values: Although some data are “statistically significant with P<0.05” in Table 2 and 4, the actual values are very helpful for readers to interpret how significant they are.
    • We thank the reviewer for this comment. We have now inserted p-values for each coefficient in table 2 and table 4. We have marked those with p-values <0.05 with bold and also provided an explanation in the footnote of the two tables.
  2. The meaning of “asterisks” in Table 4 should be displayed in the footnote.
    • The asterisks have been removed, since those with significant p-values have now been marked in bold. We hope that the reviewer agree on this approach.
  3. The “RRR” indicates “relative risk ratio” in this study, however it may be misunderstood as “relative risk reduction (1-RR)”. Simply, “RR” may be more suitable.
    • Thank you for this potential confusing description. We have now changed “RRR” to “RR” (highlighted in sections 2.6, 3.3.1, 4.1 and Table 2).
  4. During the study period (2009-2018), what could be the major or optional regimens for neo- and adjuvant therapy in this cohort?
    • Thank you for pointing this out. This has now been added to methods section 2.4 for adjuvant chemotherapy and to section 2.5.5 for neoadjuvant radiochemotherapy.
  5. Further studies are warranted to examine the oncologic impact of the observed predictors on patients’ survival. Please propose the authors’ possible future investigations.
    • Thank you for pointing this out. We have now elaborated further into perspectives of future investigations in section 4.4.

We hope that all the above correction and arguments are satisfying to the reviewer.

Round 2

Reviewer 1 Report

Comments and Suggestions for Authors

The authors have addressed my concerns.

Comments on the Quality of English Language

It's suggested to check the manuscript and correct the typo errors.